# Does the Timing of the Surgery Have a Major Role in Influencing the Outcome in Elders with Acute Subdural Hematomas?

**DOI:** 10.3390/jpm12101612

**Published:** 2022-09-30

**Authors:** Gianluca Trevisi, Alba Scerrati, Oriela Rustemi, Luca Ricciardi, Tamara Ius, Anna Maria Auricchio, Pasquale De Bonis, Alessio Albanese, Annunziato Mangiola, Rosario Maugeri, Federico Nicolosi, Carmelo Lucio Sturiale

**Affiliations:** 1Neurosurgical Unit, Ospedale Santo Spirito, Via Fonte Romana, 8, 65124 Pescara, Italy; 2Department of Neurosciences, Imaging and Clinical Sciences, G. D’Annunzio University, Via dei Vestini 31, 66100 Chieti, Italy; 3Department of Neurosurgery, S. Anna University Hospital, Via Aldo Moro, 44124 Ferrara, Italy; 4Department of Morphology, Surgery and Experimental Medicine, University of Ferrara, Via Aldo Moro, 44124 Ferrara, Italy; 5UOC Neurochirurgia 1, Azienda ULSS 8 Berica Ospedale San Bortolo, Via Giovanni Giuseppe Cappellari, 6, 36100 Vicenza, Italy; 6UOC di Neurochirurgia, Azienda Ospedaliera Sant’Andrea, Dipartimento NESMOS, Sapienza, Via di Grottarossa 1035-1039, 00189 Rome, Italy; 7Institute of Neurosurgery, Università Cattolica del Sacro Cuore, Largo Agostino Gemelli 8, 00168 Rome, Italy; 8Department of Neurosurgery, Fondazione Policlinico Universitario A. Gemelli IRCSS, Largo Agostino Gemelli 8, 00168 Rome, Italy; 9Neurosurgical Clinic AOUP “Paolo Giaccone”, Post Graduate Residency Program in Neurologic Surgery, Department of Biomedicine Neurosciences and Advanced Diagnostics, School of Medicine, University of Palermo, 90127 Palermo, Italy; 10Department of Medicine and Surgery, Division of Neurosurgery, University of Milano-Bicocca, 20126 Milan, Italy

**Keywords:** ASDH, brain, hemmorrage, surgical timing, elderly, neurosurgery

## Abstract

Background: The incidence of traumatic acute subdural hematomas (ASDH) in the elderly is increasing. Despite surgical evacuation, these patients have poor survival and low rate of functional outcome, and surgical timing plays no clear role as a predictor. We investigated whether the timing of surgery had a major role in influencing the outcome in these patients. Methods: We retrospectively retrieved clinical and radiological data of all patients ≥70 years operated on for post-traumatic ASDH in a 3 year period in five Italian hospitals. Patients were divided into three surgical timing groups from hospital arrival: *ultra-early* (within 6 h); *early* (6–24 h); and *delayed* (after 24 h). Outcome was measured at discharge using two endpoints: survival (alive/dead) and functional outcome at the Glasgow Outcome Scale (GOS). Univariate and multivariate predictor models were constructed. Results: We included 136 patients. About 33% died as a result of the consequences of ASDH and among the survivors, only 24% were in good functional outcome at discharge. Surgical timing groups appeared different according to presenting the Glasgow Outcome Scale (GCS), which was on average lower in the *ultra-early surgery* group, and radiological findings, which appeared worse in the same group. Delayed surgery was more frequent in patients with subacute clinical deterioration. Surgical timing appeared to be neither associated with survival nor with functional outcome, also after stratification for preoperative GCS. Preoperative midline shift was the strongest outcome predictor. Conclusions: An earlier surgery was offered to patients with worse clinical-radiological findings. Additionally, after stratification for GCS, it was not associated with better outcome. Among the radiological markers, preoperative midline shift was the strongest outcome predictor.

## 1. Introduction

With the increase in the mean age of the world population, the incidence of head injury and acute subdural hematoma (ASDH) in the elderly is simultaneously raising [1]. Traumatic ASDH in elderly patients is generally associated with a poor outcome [2,3]. Current guidelines on optimal treatment of these patients are based on weak evidences and, despite being considered a life-saving procedure, the role of surgery remains debated [4,5,6,7,8]. Thus, in these patients, it is quite common among clinicians preferring, at first glance, a conservative treatment, and shift toward surgery only in the case of deterioration in the state of consciousness [9].

In fact, there is controversy surrounding the appearance of the role of surgical timing in influencing the outcome. Indeed, despite, in general, a prompt surgical evacuation is regarded as an important variable associated with outcome in the natural history of ASDH [10,11,12], but this has not been confirmed by the few studies focusing on elderly patients [13,14] as the advantages of hematoma evacuation have to be balanced with the increased risk of surgery in this age range.

The aim of this study was to investigate whether the timing of surgery had a major role in influencing the outcome in a multicentric series of patients ≥70 years-old operated on for post-traumatic ASDH.

## 2. Methods

### 2.1. Population and Treatment

We selected all patients ≥70 years operated on for a post-traumatic ASDH from the retrospective databases of five Italian tertiary referral hospitals between 1 January 2017 and 31 December 2019. Non-traumatic or acute on chronic subdural hematomas were excluded as well as patients operated after hematoma chronicization. Patients with fixed and dilated pupils were also excluded since they would have invariably presented a poor outcome [8]. Presence of other intracranial post-traumatic lesions associated with ASDH was instead not an exclusion criteria, unless their role was deemed as decisive for patient neurological status, overwhelming the role of the subdural collection.

In general, indications for ASDH evacuation were based on impaired consciousness, focal neurological symptoms, hematoma thickness >1 cm, and midline shift >5 mm. As no guidelines are yet available on the appropriate treatment in the elderly, decision making and timing of the correct management were based on a case by case evaluation of the clinical status, radiological features, and family consultation.

### 2.2. Clinical and Radiological Data Collection

For each patient, we retrieved the age, sex, Charlson Comorbidity Index (CCI), history of arterial hypertension, use of antithrombotic drugs, the need for an urgent coagulopathy correction at A&E admission, mechanism of injury, neurological status measured by the Glasgow Coma Scale (GCS) at admission and during the entire preoperative period, pupillary size and light reaction, neurological deficits, and seizures. Patients were divided in three GCS level groups, both at arrival and soon before surgery: mild (13–15), moderate (9–12), and severe (3–8).

The radiological parameters were retrieved by local PACS. For each patient, we collected the ASDH thickness, midline shift, and the presence of other post-traumatic lesions at the first CT-scan, at any CT-scan performed in the preoperative period, and at the first post-operative CT-scan (within 24 h from surgery). The ASDH thickness/midline shift ratio was computed in all cases.

### 2.3. Timing of Surgery Groups

The timing of surgery was calculated from A&E arrival to the starting time of the surgical procedure as recorded on the surgical reports. Patients were divided into three groups according to the timing of surgery:(1)Ultra-early, within 6 h from A&E arrival;(2)Early, between 6 h and 24 h form arrival;(3)Delayed, after 24 h.

Timing was chosen by the surgical team in a case-by-case fashion according to the clinical status, radiological evidence, the presence of comorbidities, need for coagulopathy reversal, patients and families wills, etc.

### 2.4. Outcome Measures

Outcome was measured at discharge. We used two main endpoints: (1) survival (alive or dead); and (2) functional outcome (good or poor).

The functional outcome was measured according to the Glasgow Outcome Scale (GOS) as death (D), vegetative status (VS), severe disability (SD), moderate disability (MD), and good recovery (GR). GOS 1–3 (D, VS, SD) were considered as a poor outcome and 4–5 (MD, GR) as a good outcome. 

### 2.5. Statistical Analysis

ANOVA test (for quantitative variables), Chi-squared test (for qualitative variables), and their respective post hoc tests with Bonferroni’s correction were used to compare the differences in the clinical-radiological characteristics, survival, and survival among the three groups with different surgical timing. Outcomes were also stratified by preoperative GCS level.

Logistic regression models were used to assess the association of survival and functional outcome with age, GCS at arrival and immediately preoperative GCS, first CT and last preoperative ASDH thickness, first CT and last preoperative midline shift, and with the surgical timing groups.

Statistical analysis was performed using JASP, an R based software developed by the JASP team University of Amsterdam (Amsterdam—The Netherlands), Version 0.16.2; significance was set at *p* < 0.05.

## 3. Results

### 3.1. Demographics, Radiological, and Clinical Data

The demographical, radiological, and clinical data of the included patients are reported in Table 1. 

We included 136 patients ≥ 70 years operated on for a traumatic ASDH along a 3-year time span who met our inclusion criteria. The mean age was 78.5 ± 5.7 years (Min–Max: 70–92) and 76 (56%) patients were males. Mean CCI was 5.3 ± 1.7, with 84 patients (61.8%) under antithrombotic drugs.

We divided patients according to three surgical timing groups: 30 (22%) underwent ultra-early surgery; 76 (56%) had early surgery, and 30 (22%) had delayed surgery.

With regard to the timing of surgery, in general, a worsening GCS level corresponded with an earlier surgical procedure. Patients undergoing ultra-early surgery were those who arrived in a significantly worse clinical status (lower GCS) and also showed a more severe neuroradiological picture at the first CT-scan, namely, a thicker ASDH and wider midline shift compared with patients undergoing surgery after 6 h from admission. In fact, the post hoc analysis showed a significant difference between ultra-early and delayed surgery groups in the mean GCS at presentation and at first CT-scan parameters.

Some patients, instead, showed a worsening clinical and/or radiological picture during their stay in the emergency department, leading to a shift from a first conservative approach to a surgical indication. Indeed, preoperative GCS showed no significant difference among the different surgical timing groups.

From a radiological point of view, 43.3% of patients included in the delayed surgery group showed worsening neuroimaging parameters at the last preoperative CT-scan compared with the first one, with a significant difference when compared with the other groups (*p* < 0.001). In particular, the midline shift observed in the delayed surgery group showed a mean increase of 3 mm, which appeared significant when compared with that of 0.25 mm observed in the ultra-early surgery group and 1.8 mm in the early surgery group.

In summary, we did not observe significant differences in the GCS status, hematoma thickness, and preoperative midline shift among the three surgical timing groups.

A significant difference was instead seen in the duration of surgery, which appeared usually longer in the early surgery group.

Primary decompressive craniectomy (DC) was performed in 11 cases: three ultra-early, six early, and two delayed surgery, respectively. Three early-surgery cases underwent secondary DC due to post-operative brain swelling.

Regarding complications in both surgery related ones (rebleeding, stroke, new-onset seizures or deficits, wound problems) and systemic complications (cardio-pulmonary, systemic infection, wound problems), we did not observe significant differences among the surgical timing groups. In particular, rebleeding occurred in 11 cases: three in ultra-early, seven in early, and one in the delayed surgical timing group, respectively. Three were on anticoagulation and four on antiplatelets, while four patients were not on antithrombotics. Reoperation was performed in three of these cases. All rebleeding patients had a poor outcome at discharge: seven died, two were in vegetative state, and two had a severe disability.

### 3.2. Relationship between Surgical Timing and Clinical Outcome

Overall, 91 patients (67%) were alive at discharge, but only 33 (24% of the total number of patients) appeared to have a good functional outcome.

Surgical timing did not appear to be neither associated with survival nor with good functional outcome; on the other hand, we observed a trend of association with worse outcome in the ultra-early surgery group (Table 2).

The stratification of outcomes according to the preoperative GCS classes showed a significant higher rate of mortality in patients with mild GCS undergoing ultra-early surgery compared with the other groups. However, these patients were significantly older than those showing that mild GCS operated later than 6 h from diagnosis. Moreover, this subgroup of patients also showed a larger midline shift at both the first and last preoperative CT-scan compared with the other patients operated in mild GCS, even though these did not reach a statistically significant level.

Preoperative midline shift appeared as the only covariate associated with survival and functional outcome in the logistic regression analysis (Table 3).

## 4. Discussion

Surgical evacuation represents the only possible life-saving treatment in severely symptomatic ASDH. However, survival and good functional outcome remain poor in many elderly patients, despite an aggressive treatment [14,15,16,17]. Therefore, in this age range, the treatment of choice is still debated and the object of ongoing trials [6,7], and most neurosurgeons prefer an initial conservative treatment, opting for surgery only in the case of an impaired state of consciousness. Although, while surgery has also been shown to have a major life-saving role in these patients when presenting with a severe clinical status, several studies have reported a high rate of poor functional outcome both at discharge and at 6 months follow-up, often due to the higher incidence of perioperative complications [8].

Several surgical techniques have been described to evacuate an ASDH. The most commonly used being craniotomy (bone is repositioned after hematoma evacuation) and craniectomy (a decompressive procedure where the dura is expanded and bone is not re-placed at the end of surgery, needing a subsequent cranioplasty surgery), with no clear superiority of any of the two in terms of post-operative outcome [18,19]. However, recently, other techniques aimed to minimize surgical trauma have been proposed such as endoscopic/endoscope-assisted evacuation [20] and minicraniotomy in elderly patients [21,22,23,24]. Altogether, regardless of the surgical technique, the results largely depend on the clinical/radiological features and there is not a standardized indication to select those patients who will maximally benefit from a specific technique.

The timing of surgery has often been considered as an important variable associated with outcome in ASDH evacuation [10,11,12]. However, several studies have failed to demonstrate a strict association between the timing of surgery and outcome [25,26,27] and only a few of them have specifically focused on the elderly, also failing to show an advantage of early aggressive management compared with delayed surgery in these patients [13,14]. Our study also seems to confirm these findings, but it is always necessary to bear in mind that different surgical timings generally reflect different clinical severities at onset [10,28]. In agreement, our data additionally showed that in this age range, the leading indication for surgery was lower GCS and higher ASDH thickness with midline shift, which represented the strongest motivation for indicating surgery earlier than for other categories. Interestingly, on the other hand, age, general comorbidities, and antithrombotic drug assumption did not appear to influence the timing of surgery.

Similarly, a different category of patients characterized by an apparent mismatch between the radiological picture (considered as alarming for impending clinical worsening) and a mild GCS status at onset were usually earlier operated on despite an older age (Table 2). Nonetheless, these patients (overall 5) showed a worse outcome compared to those with similar GCS undergoing surgery with a delayed timing. Possible reasons for this apparently discordant result can be the radiological criteria leading to ultra-early surgery and the more advanced age of these five patients.

In our series, the main outcome predictor for the survival and functional outcome was the entity of the preoperative midline shift, which is usually, together with GCS, the main prognostic factor in elderly patients with ASDH [29].

The detailed analysis of the GCS trend and radiological parameters showed that clinicians used less strict criteria to indicate surgery in elderly patients with an ASDH with a mean ASDH thickness >1 cm and a mean midline shift > 5 mm at the first CT-scan in those patients undergoing a delayed surgery. In addition, the mean presenting GCS in these delayed surgery patients was rather low (12.4 ± 3.5) compared to what was expected in ASDH cases undergoing surgery. This is probably due to the awareness of the intrinsic high risk of this surgery in elderly patients [8,17,29,30,31], thus reflecting the current uncertainty of the best initial treatment in these patients [7]. Furthermore, this reflects the tendency of most neurosurgeons to choose to delay a possible craniotomy in elderly patients who do not show an initial serious clinical condition in the hope of a secondary chronicization of the hematoma, opening the possibility of its evacuation with a minimally invasive technique [32,33]. However, all the patients included in this case series were operated during the acute phase of the hematoma.

### Study Limitations

The main limitations of our study are its retrospective nature and the possible heterogeneous indications for surgery that influenced its timing among the different centers as well as among different surgeons in the same hospital.

We decided to exclude those patients presenting with fixed and dilated pupils who underwent surgery from the analysis. Indeed, in line with the evidence of other authors, we had previously shown that while these patients were operated in an ultra-early timing, they invariably had a poor outcome [8]. This is due to the presenting status and not to the timing of the surgery, which was calculated from hospital arrival and not from trauma, which cannot be influenced by the neurosurgical team and should be the object of a different discussion. We decided to exclude these patients to avoid a possible bias in the assessment of the results of ultra-early surgery.

In general, ultra-early surgery was reserved to patients with worse clinical and radiological findings at arrival, which is known to strongly influence the outcome [17,28,29,34]. Moreover, the clinical and radiological scenarios of the ASDH patients are dynamic and heterogeneous, with a number of patients who will maintain stable conditions and others who will suffer a worsening, which will eventually influence the timing of the surgery and post-operative outcome [35]. To partially overcome this limitation, we stratified patients according to their preoperative GCS (Table 2), with no substantial difference among the surgical timing groups. This reinforces the concept that clinical/radiological findings are critical in determining the outcome and that we should avoid waiting to operate until patients get to a critical condition. Indeed, it is not the intention of the present study to convey the message to delay surgery in critical cases. This series depicts our real world practice, where critical patients more often underwent ultra-early surgery.

Moreover, the low number of patients in each group may have limited the statistical power preventing some variables such as the GCS and ASDH thickness to reach a significance.

Additionally, due to the retrospective multicentric nature of the study, post-operative management was not set according to a standard protocol, as any participant center may have customized some details. These could have concurred in influencing the outcomes. However, all of the centers followed the standard clinical practice and guidelines in terms of ICP and blood pressure management as well as in terms of VTE and seizure prophylaxis. Therefore, we believe that post-operative management could have only marginally influenced the outcomes in the present series.

A prospective trial where patients with similar preoperative clinical and radiological characteristics, for whom different timing of surgery could be appropriate, randomized among different surgical timing groups, and following the same post-operative management, could further clarify the influence of surgical timing on the outcomes.

Finally, we did not take into account the occurrence of concomitant brain contusions in influencing the final outcome, even though we did not observe in any case a significant contusion growth needing intervention [36].

## 5. Conclusions

In patients ≥70 years old, operated on for a post-traumatic ASDH, the main factors associated with the timing of surgery were GCS and radiological findings such as ASDH thickness and midline shift, with preoperative midline shift emerging as the only factor associated with survival and functional outcome at the multivariate analysis. The timing of surgery neither influenced the survival nor functional outcome. However, critical patients were almost always treated in an ultra-early timing.

## Figures and Tables

**Table 1 jpm-12-01612-t001:** Demographical, radiological and clinical data.

	Ultra-Early (<6 h) n = 30 (22%)	Early (6–24 h) n = 76 (56%)	Delayed (>24 h) n = 30 (22%)	Total n = 136	*p*
Age in years	78.8 ± 5.7	77.9 ± 5.8	79.9 ± 5.3	78.5 ± 5.7	*ns*
Male gender	13 (43.3%)	46 (60.5%)	17 (56.7%)	76 (56%)	*ns*
Left hemisphere	11 (36.7%)	40 (52.6%)	17 (56.7%)	68 (50%)	*ns*
Charlson Comorbidity Index	5.4 ± 1.8	5.1 ± 1.7	5.6 ± 1.9	5.3 ± 1.7	*ns*
Use of antithrombotic drugs	16 (53.3%)	47 (61.8%)	21 (70%)	84 (61.8%)	*ns*
GCS level at A&E arrival:					**0.001**
Mild	9 (30%)	24 (31.6%)	20 (66.7%)	53 (39%)	**0.002**
Moderate	5 (16.7%)	23 (30.3%)	7 (23.3%)	35 (25.7%)	*ns*
Severe	16 (53.3%)	29 (38.2%)	3 (10%)	48 (35.3%)	**0.002**
Mean GCS at A&E arrival	8.5 ± 4.6 *	10.2 ± 4.3	12.4 ± 3.5 *	10.3 ± 4.4	**0.006**
Preoperative drop of GCS	6 (20%)	16 (21.1%)	12 (40%)	34 (25%)	**0.09**
Preoperative GCS level:					**0.04**
Mild	5 (16.7%)	15 (19.7%)	13 (43.3%)	33 (24.3%)	**0.02**
Moderate	6 (20%)	22 (28.9%)	8 (26.7%)	36 (26.5%)	*ns*
Severe	19 (63.3%)	39 (51.3%)	9 (30%)	67 (49.3%)	**0.03**
Mean preoperative GCS	7.3 ± 3.9	8.7 ± 4.2	10 ± 4.3	8.7 ± 4.2	*ns*
Anisocoric pupils	12 (44.4%)	19 (29.2%)	6 (20%)	37 (30.3%)	*ns*
Neurological deficits or epilepsy	16 (53.3%)	36 (47.4%)	19 (63.3%)	71 (52.2%)	*ns*
Associated post-traumatic lesions	13 (43.3%)	31 (40.8%)	13 (43.3%)	57 (41.9%)	*ns*
First CT ASDH thickness in mm	17.5 ± 5.9 *	15.4 ± 6.4	12.4 ± 7.8 *	15.2 ± 6.8	**0.02**
First CT midline shift in mm	11.5 ± 6.8 *	8.5 ± 5.7	5.6 ± 4.2 *	8.5 ± 6	**<0.001**
First CT thickness/midline shift ratio	2.4 ± 2.9	2.8 ± 3.7	2.4 ± 1.4	2.6 ± 3.1	*ns*
Preoperative radiological worsening	2 (6.7%)	12 (15.8%)	13 (43.3%)	27 (19.8%)	**<0.001**
ASDH thickness variation (first CT—preoperative CT) in mm	0.7 ± 2.8	1.7 ± 4.6	3 ± 5.8	1.8 ± 4.6	*ns*
Preoperative ASDH thickness in mm	18.2 ± 5.8	17.2 ± 5.7	15.5 ± 7.2	17 ± 6.1	*ns*
Midline shift variation (first CT—preoperative CT) in mm	0.25 ± 0.9 *	1.8 ± 4.4	3 ± 5.1 *	1.7 ± 4.1	**0.04**
Preoperative midline shift in mm	11.8 ± 6.5	10.3 ± 5.5	8.6 ± 4.9	10.3 ± 5.7	*ns*
Preoperative thickness/midline shift ratio	2.1 ± 1.4	2.6 ± 3.5	2.0 ± 1.3	2.3 ± 2.7	*ns*
Craniotomy size in cm^2^	44.3 ± 29.1	41.6 ± 31.8	33.2 ± 26.9	40.3 ± 30.2	*ns*
Duration of surgery in minutes	114 ± 35	155 ± 53 *	102 ± 47 *	133 ± 54	**0.002**
Post-operative midline shift in mm	5.4 ± 6.1	4.3 ± 5.6	2.5 ± 4	4.1 ± 5.5	*ns*
Length of hospital stay in days	21.9 ± 23.5	17.3 ± 21.6	16.1 ± 12.4	18 ± 20.4	*ns*

**Legend:** All figures express mean ± standard deviation or frequency (percentage) of data referred to the corresponding column. * Significative difference between groups at post-hoc test. Abbreviations: GCS: Glasgow Coma Scale; A&E: Accident and Emergency Department; CT: Computed-Tomography; ASDH: acute subdural hematoma.

**Table 2 jpm-12-01612-t002:** Demographics and outcomes stratified by preoperative GCS.

Preoperative GCS (Total n = 136)			Surgical Timing			*p*
			Ultra-Early (within 6 h) n = 30 (22%)	Early (6–24 h) n = 76 (60%)	Delayed (after 24 h) n = 30 (22%)	
**Overall**	**Survival**	**Alive** (n = 91; 70%) **Dead** (n = 45; 45%)	16 (53.3%)	55 (72.4%)	20 (66.7%)	*ns*
			14 (46.7%)	21 (27.6%)	10 (33.3%)	
	**Functional Outcome**	**Good** (n = 33; 24%) **Poor** (n = 103; 76%)	5 (16.7%)	18 (23.7%)	10 (33.3%)	*ns*
			25 (83.3%)	58 (76.3%)	20 (66.7%)	
**Mild**	**Total (n = 33; 24.2%)**		**5 (16.7%)**	**15 (19.7%)**	**13 (43.3%)**	**0.02**
	Mean Age		84.2 ± 4.5 *	77.7 ± 4.5 *	79.8 ± 5	**0.04**
	Mean GCS at A&E arrival		13.75 ± 0.9	14.3 ± 0.7	14 ± 1.6	*ns*
	Mean first CT ASDH thickness in mm		15 ± 8	13.9 ± 3.4	13.8 ± 9.5	*ns*
	Mean first CT midline shift in mm		8.8 ± 4.8	5.4 ± 3.8	4.7 ± 3.5	*ns*
	Mean preoperative GCS		13.75 ± 0.9	14.3 ± 0.7	14.4 ± 0.7	*ns*
	Mean preoperative ASDH thickness in mm		17.8 ± 8.2	15.8 ± 4.5	14.9 ± 8.5	*ns*
	Mean preoperative midline shift in mm		9.6 ± 4.2	9.2 ± 5.7	5.8 ± 2.6	*ns*
	**Survival**	**Alive** (n = 25; 75.8%)	1 (20%)	13 (86.7%)	11 (84.6%)	**0.007**
		**Dead** (n = 8; 24.2%)	4 (80%)	2 (13.3%)	2 (15.4%)	
	**Functional Outcome**	**Good** (n = 14; 42.4%)	1 (20%)	8 (54.3%)	5 (38.5%)	*ns*
		**Poor** (n = 19; 57.6%)	4 (80%)	7 (46.7%)	8 (61.5%)	
**Moderate**	**Total (n = 36; 26.5%)**		**6 (20%)**	**22 (28.9%)**	**8 (26.7%)**	*ns*
	Mean Age		78 ± 4.7	77 ± 6.4	78.9 ± 5	*ns*
	Mean GCS at A&E arrival		11.8 ± 1.9	11.2 ± 1.4	12.9 ± 2.3	*ns*
	Mean first CT ASDH thickness in mm		13.2 ± 6.3	16.1 ± 7.7	10.3 ± 5.2	*ns*
	Mean first CT midline shift in mm		7.4 ± 4.3	7.8 ± 4.9	4.8 ± 3.9	*ns*
	Mean preoperative GCS		10.6 ± 0.9	10.4 ± 1	10.3 ± 1.1	*ns*
	Mean preoperative ASDH thickness in mm		13.2 ± 6.3	17.9 ± 5.4	15.7 ± 4.4	*ns*
	Mean preoperative midline shift in mm		7.4 ± 4.3	8.5 ± 3.9	12.4 ± 6.4	*ns*
	**Survival**	**Alive** (n = 28; 77.8%)	4 (66.7%)	19 (86.4%)	5 (62.5%)	*ns*
		**Dead** (n = 8; 22.2%)	2 (33.3%)	3 (13.6%)	3 (37.5%)	
	**Functional Outcome**	**Good** (n = 12; 33.3%)	3 (50%)	6 (27.3%)	3 (37.5%)	*ns*
		**Poor** (n = 24; 67.7%)	3 (50%)	16 (72.7%)	5 (62.5%)	
**Severe**	**Total (n = 67; 49.3%)**		**19 (63.3%)**	**39 (51.3%)**	**9 (30%)**	**0.03**
	Mean Age		77.6 ± 5.6	78.4 ± 5.9	80.9 ± 6	*ns*
	Mean GCS at A&E arrival		6.4 ± 4.2	7.6 ± 4.6	10.2 ± 4.8	*ns*
	Mean first CT ASDH thickness in mm		19.3 ± 4.4	15.8 ± 6.8	12.3 ± 7.2	*ns*
	Mean first CT midline shift in mm		13.4 ± 7.2	10.4 ± 6.3	7.2 ± 5.3	*ns*
	Mean preoperative GCS		4.9 ± 1.9	5 ± 2.1	4.9 ± 1.6	*ns*
	Mean preoperative ASDH thickness in mm		19.7 ± 4.3	17.5 ± 6.4	16 ± 7.7	*ns*
	Mean preoperative midline shift in mm		13.6 ± 7	12 ± 5.8	9.4 ± 4	*ns*
	**Survival**	**Alive** (n = 38; 56.7%)	11 (57.9%)	23 (59%)	4 (44%)	*ns*
		**Dead** (n = 29; 43.3%)	8 (42.1%)	16 (41%)	5 (55.6%)	
	**Functional Outcome**	**Good** (n = 7; 10.4%)	1 (5.3%)	4 (10.3%)	2 (22.2%)	*ns*
		**Poor** (n = 60; 89.6%)	18 (94.7%)	35 (89.7%)	7 (77.8%)	

**Legend:** All figures express mean ± standard deviation or frequency (percentage) of data referred to the corresponding column. * Significative difference between groups at post-hoc test. Functional Outcome classified according to Glasgow Outcome Scale: “Good” = Good Recovery/Moderate Disability; “Poor” = Severe Disability/Vegetative State/Death. Abbreviations: GCS: Glasgow Coma Scale; A&E: Accident and Emergency Department; CT: Computed-Tomography; ASDH: acute subdural hematom.

**Table 3 jpm-12-01612-t003:** Binomial logistic regression: analysis of variables associated to survival and functional outcome.

Dependent Variable	AUC	Covariates	Odds Ratio	*p*-Value	95% Confidence Interval	
					Lower Bound	Upper Bound
**Survival** (“Dead” coded as class 1)	0.746	Age	1.060	0.204	−0.032	0.148
		GCS at A&E arrival	0.962	0.653	−0.206	0.129
		First CT ASDH thickness (mm)	1.075	0.498	−0.137	0.282
		First CT midline shift (mm)	0.878	0.284	−0.368	0.108
		Preoperative GCS	0.954	0.566	−0.207	0.113
		Preoperative ASDH thickness (mm)	0.895	0.261	−0.305	0.083
		Preoperative midline shift (mm)	1.230	**0.044 ***	0.005	0.409
		Surgical timing (Ultra-Early)	0.016	0.237	−11.071	2.738
		Surgical timing (Early)	0.006	0.138	−12.004	1.662
		Surgical timing (Delayed)	0.008	0.174	−11.911	2.149
**Functional Outcome** (“Poor” coded as class 1)	0.805	Age	1.034	0.498	−0.063	0.130
		GCS at A&E arrival	0.884	0.237	−0.327	0.081
		First CT ASDH thickness (mm)	1.123	0.289	−0.098	0.329
		First CT midline shift (mm)	0.902	0.482	−0.389	0.184
		Preoperative GCS	0.902	0.184	−0.255	0.049
		Preoperative ASDH thickness (mm)	0.885	0.228	−0.320	0.076
		Preoperative midline shift (mm)	1.282	**0.038 ***	0.013	0.483
		Surgical timing (Ultra-Early)	1.309	0.946	−7.509	8.048
		Surgical timing (Early)	0.414	0.819	−8.428	6.664
		Surgical timing (Delayed)	0.553	0.880	−8.267	7.083

**Legend:** * Statistically significant. Functional Outcome classified according to Glasgow Outcome Scale: “Good” = Good Recovery/Moderate Disability; “Poor” = Severe Disability/Vegetative State/Death. Abbreviations: AUC: Area Under the Curve; GCS: Glasgow Coma Scale; A&E: Accident and Emergency Department; CT: Computed-Tomography; ASDH: acute subdural hematoma.

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
