# Peer review of "Does the Timing of the Surgery Have a Major Role in Influencing the Outcome in Elders with Acute Subdural Hematomas?"

_jpm, 2022, doi:10.3390/jpm12101612_

Round 1

Reviewer 1 Report

The manuscript "Has the timing of the surgery a major role in influencing outcome in elderly patients with acute subdural hematomas?" addresses an important issue: the use of surgical evacuation as the only possible life-saving treatment in severely symptomatic ASDH and whether the timing of surgery influences patient’s outcome. Despite the aggressive treatment, survival and good functional outcome remain poor in many elderly patients. The authors investigated if timing of surgery had a major role in influencing outcome in a multicentric series of patients ≥ 70 years-old operated for a post-traumatic ASDH in a 3 years period among 5 Italian hospitals.

The objectives were clearly stated and explained in the manuscript, however the experimental strategy raises some major concerns and so the experimental information from which the conclusions were drawn. The manuscript is overall well written and has good organization with minor English language and style spell check required. The authors have done a great job on analyzing the experimental data and on discussing the results and their limitations, considering always different alternative explanations/considerations for interpreting the results.

The paper is interesting but there is a need for more experimental detail in order to critically review the data. Specifically, they should provide information for the following questions and comments:

Major points:

1.     It is suggested to carry out a more current review of the literature incorporating bibliography in the Introduction as well as in the Results sections. Specially focusing on other appropriate treatment of ASDHs and how do they compare with each other. The authors may also discuss this topic more extensively in the Discussion section.

2.     Major concerns arise form the significance of the sample chosen. Although the authors honestly discussed this issue in the manuscript, I would suggest extending the sample to other hospitals and a larger number of patients.

3.     How does the exclusion of some patients likely presenting a poor outcome (for example Patients with fixed and dilated pupils) from the analyses affect the final results? A paragraph analyzing the whole sample including those patients may be added to the Main text or the Supplementary Material section.

Minor points:

1.     Abbreviation of GCS does not appear in the abstract.

2.     In line 50-52 the sentence needs to be re-written and/or English checked as it is not comprehensive as is in the current state of the manuscript.

3.     Sub-section titles should be highlighted from the plain text encompassed in each sub-section, e.g. italicized, as in lines 59, 74, etc. throughout the entire manuscript.

4.     In the first paragraph of the “3. Results” section symbols are missing, this issue is repeated through the manuscript at different sections (e.g. line 260, line 306 and so on).

5.     Superscript is missing in some references at some points: line 255, line 263…

Author Response

The manuscript "Has the timing of the surgery a major role in influencing outcome in elderly patients with acute subdural hematomas?" addresses an important issue: the use of surgical evacuation as the only possible life-saving treatment in severely symptomatic ASDH and whether the timing of surgery influences patient’s outcome. Despite the aggressive treatment, survival and good functional outcome remain poor in many elderly patients. The authors investigated if timing of surgery had a major role in influencing outcome in a multicentric series of patients ≥ 70 years-old operated for a post-traumatic ASDH in a 3 years period among 5 Italian hospitals.

The objectives were clearly stated and explained in the manuscript, however the experimental strategy raises some major concerns and so the experimental information from which the conclusions were drawn. The manuscript is overall well written and has good organization with minor English language and style spell check required. The authors have done a great job on analyzing the experimental data and on discussing the results and their limitations, considering always different alternative explanations/considerations for interpreting the results.

The paper is interesting but there is a need for more experimental detail in order to critically review the data. Specifically, they should provide information for the following questions and comments:

Reviewer question:

Major points:

  1. It is suggested to carry out a more current review of the literature incorporating bibliography in the Introduction as well as in the Results sections. Specially focusing on other appropriate treatment of ASDHs and how do they compare with each other. The authors may also discuss this topic more extensively in the Discussion section.

Authors answer:

We included a more extensive discussion on surgical option in the discussion section.

Reviewer question:

  1. Major concerns arise form the significance of the sample chosen. Although the authors honestly discussed this issue in the manuscript, I would suggest extending the sample to other hospitals and a larger number of patients.

Authors answer:

While Authors agree that a larger sample would increase the power of the study, unfortunately this is not feasible due to the significant time constraints given for review (8 days).

Reviewer question:

  1. How does the exclusion of some patients likely presenting a poor outcome (for example Patients with fixed and dilated pupils) from the analyses affect the final results? A paragraph analyzing the whole sample including those patients may be added to the Main text or the Supplementary Material section.

Authors answer:

All patients who arrived with fixed and dilated pupils died despite ultra-early surgery. In Authors’ opinion this is due to the presenting status and not to timing of surgery, which was calculated from hospital arrival and not from trauma, which cannot be influenced by the neurosurgical team and should be object of a different discussion. We decided to exclude this patients to avoid a possible bias in the assessment of the results of ultra-early surgery.

We included the above in the limitations sub-heading of the discussion section.

Reviewer question:

Minor points:

  1. Abbreviation of GCS does not appear in the abstract.

Authors answer:

Amended

Reviewer question:

  1. In line 50-52 the sentence needs to be re-written and/or English checked as it is not comprehensive as is in the current state of the manuscript.

Authors answer:

Reformulated

Reviewer question:

  1. Sub-section titles should be highlighted from the plain text encompassed in each sub-section, e.g. italicized, as in lines 59, 74, etc. throughout the entire manuscript.

Authors answer:

Amended

Reviewer question:

  1. In the first paragraph of the “3. Results” section symbols are missing, this issue is repeated through the manuscript at different sections (e.g. line 260, line 306 and so on).

Authors answer:

Amended

Reviewer question:

  1. Superscript is missing in some references at some points: line 255, line 263…

Authors answer:

Amended

Reviewer 2 Report

Very interesting topic, well planned and exposed. Congratulations!!

.- However, some comments are made in favor of improving the current version of the manuscript.

Title. Very suggestive as a specific question. Perhaps consider changing “elderly patients” to “elders” or “inpatient elders”

Summary. Maybe specify subheading “objetives” after “background”.

Keywords. 7 words, may be many. Rate this section.

Introduction. It is noteworthy that more than a third of the citations appeared in this section. Thirteen of the 31 quotes provided appear in this section.

Methodology. Perhaps the last sentence resolved very succinctly, add that statistical significance was thought for p<0.05.

Results. V. comments of the tables.

Discussion. Study limitations. This subsection may be long.

Conclusions. The first sentence describes the results. It should be considered changing the beginning of this section. Very long sentence. Assess redo in two or three sentences. The last sentence may be included in the section on limitations, insufficient evidence and this study with a small sample.

References. Of the 31 references provided, 18 are recent (5 years or less since their publication). Consider including some additional recent citation.

Tables. 3 tables are exposed. Perhaps Tables 1 and 2 provide too much data for the reader. Evaluate summarizing a section, presenting the most relevant results in the text or presenting the tables as they are as complementary material.

Author Response

Comments and Suggestions for Authors

Very interesting topic, well planned and exposed. Congratulations!!

However, some comments are made in favor of improving the current version of the manuscript.

Reviewer question:

Title. Very suggestive as a specific question. Perhaps consider changing “elderly patients” to “elders” or “inpatient elders”

Authors answer:

Thank you, we changed the title as suggested

Reviewer question:

Summary. Maybe specify subheading “objetives” after “background”.

Authors answer:

For Abstract sections we followed  the Journal’s Instructions for Authors

Reviewer question:

Keywords. 7 words, may be many. Rate this section.

Authors answer:

Also for this we followed the Instructions for Authors “three to ten pertinent keywords need to be added after the abstract”

Reviewer question:

Introduction. It is noteworthy that more than a third of the citations appeared in this section. Thirteen of the 31 quotes provided appear in this section.

Authors answer:

We enriched the discussion section, including new citations.

Reviewer question:

Methodology. Perhaps the last sentence resolved very succinctly, add that statistical significance was thought for p<0.05.

Authors answer:

Amended

Reviewer question:

Results. V. comments of the tables.

Authors answer:

Please refer to response on tables’ related comments.

Reviewer question:

Discussion. Study limitations. This subsection may be long.

Authors answer:

We believe that a fair statement of study limitations is extremely important, especially in retrospective manuscripts, to avoid improper generalizations of study findings.

Reviewer question:

Conclusions. The first sentence describes the results. It should be considered changing the beginning of this section. Very long sentence. Assess redo in two or three sentences. The last sentence may be included in the section on limitations, insufficient evidence and this study with a small sample.

Authors answer:

We corrected the first sentence and deleted the last one, already included in the limitations section

Reviewer question:

References. Of the 31 references provided, 18 are recent (5 years or less since their publication). Consider including some additional recent citation.

Authors answer:

We added some recent citations.

Reviewer question:

Tables. 3 tables are exposed. Perhaps Tables 1 and 2 provide too much data for the reader. Evaluate summarizing a section, presenting the most relevant results in the text or presenting the tables as they are as complementary material.

Authors answer:

We amended the tables

Round 2

Reviewer 1 Report

The authors have answered the questions and comments proposed in a fruitful manner, thus I consider the article may be published in the current version.